# Immunomodulation by Hemoadsorption—Changes in Hepatic Biotransformation Capacity in Sepsis and Septic Shock: A Prospective Study

**DOI:** 10.3390/biomedicines10102340

**Published:** 2022-09-20

**Authors:** Janina Praxenthaler, Elke Schwier, Simon Altmann, Carmen Kirchner, Julian Bialas, Dietrich Henzler, Thomas Köhler

**Affiliations:** 1Department of Anesthesiology, Surgical Intensive Care, Emergency and Pain Medicine, Ruhr-University Bochum, Klinikum Herford, 32049 Herford, Germany; 2Department of Anesthesiology, Intensive Care and Pain Medicine, KSOB-Clinics, Klinikum Traunstein, 83278 Traunstein, Germany; 3Department of Anesthesiology, Surgical Intensive Care, Emergency and Pain Medicine, Ruhr University Bochum, Knappschaftskrankenhaus Bochum, 44892 Bochum, Germany; 4Department of General and Visceral Surgery, Thoracic Surgery and Proctology, Ruhr University Bochum, Klinikum Herford, 32049 Herford, Germany; 5Department of Data Science and Intelligent Analytics, University of Applied Science, FH Kufstein Tirol, 6330 Kufstein, Austria; 6Department of Anesthesiology and Intensive Care Medicine, AMEOS-Klinikum Halberstadt, 38820 Halberstadt, Germany

**Keywords:** CytoSorb^®^, hemoadsorption, septic shock, sepsis, hepatic dysfunction, sepsis-associated liver dysfunction, LiMAx^®^, static liver parameters, dynamic liver function test, Interleukin-6

## Abstract

Background: Sepsis is often associated with liver dysfunction, which is an indicator of poor outcomes. Specific diagnostic tools that detect hepatic dysfunction in its early stages are scarce. So far, the immune modulatory effects of hemoadsorption with CytoSorb^®^ on liver function are unclear. Method: We assessed the hepatic function by using the dynamic LiMAx^®^ test and biochemical parameters in 21 patients with sepsis or septic shock receiving CytoSorb^®^ in a prospective, observational study. Points of measurement: T_1_: diagnosis of sepsis or septic shock; T_2_ and T_3_: 24 h and 48 h after the start of CytoSorb^®^; T_4_: 24 h after termination of CytoSorb^®^. Results: The hepatic biotransformation capacity measured by LiMAx^®^ was severely impaired in up to 95 % of patients. Despite a rapid shock reversal under CytoSorb^®^, a significant improvement in LiMAx^®^ values appeared from T_3_ to T_4_. This decline and recovery of liver function were not reflected by common parameters of hepatic metabolism that remained mostly within the normal range. Conclusions: Hepatic dysfunction can effectively and safely be diagnosed with LiMAx^®^ in ventilated ICU patients under CytoSorb^®^. Various static liver parameters are of limited use since they do not adequately reflect hepatic dysfunction and impaired hepatic metabolism.

## 1. Introduction

Sepsis is not only one of the most common diseases worldwide, but it is also a leading cause of death [1]. It is associated with about one out of five deaths globally [1,2]. Despite multiple diagnostic and therapeutic efforts in the last years, the 90-day-mortality in septic shock remains almost unchanged at a high rate of close to 40% in Europe [3]. In critically ill patients, hepatic insufficiency or liver failure is an indicator of poorer outcomes [4,5]. Especially early liver impairment is associated with an increased hospital mortality rate and occurred in 11% of seriously ill patients [6]. Even mild hyperbilirubinemia has been identified as an independent predictor of mortality in patients with sepsis [7,8,9].

Sepsis and septic shock are frequently accompanied by sepsis-associated liver dysfunction (SALD). However, determining the incidence of SALD is difficult due to the lack of a uniformly accepted definition. As a result, a comprehensive registry is missing and the exact incidence is unknown [10,11,12].

The liver is strongly implicated in the development and amplification of the multiple organ dysfunction syndrome (MODS). The massive hepatic release of cytokines and activation of multiple immunological pathways further compromise the clinical situation [13,14,15,16]. Clinically, hyperbilirubinemia is one expression of complex hepatic dysfunction and it depends on various factors, e.g., immunological functional status, current liver perfusion, individual factors, or metabolic situation [17,18,19,20,21,22].

To assess liver function in sepsis, paraclinical static parameters are mostly used, such as elevated serum concentrations of bilirubin, alkaline phosphatase, or aminotransferases. Using these parameters is currently the best approach because a gold standard or a generally available specific test is missing [9,10,12,23].

The maximum liver function capacity test (LiMAx^®^) is a non-invasive procedure and a good predictor of mortality in patients with sepsis [24]. This dynamic diagnostic tool can detect septic-related liver dysfunctions at an early stage [24,25]. The compromised hepatic biotransformation capacity correlates with Endothelin-1 and IL-6 in severely ill patients [24,26]. Additionally, the LiMAx^®^ test has been successfully used in the ICU for monitoring antibiotics and for dosing antimycotics [27,28,29]. It seems to be superior to static liver parameters [24,30,31].

The LiMAx^®^ test monitors liver function based on a ^13^C-breath test. Intravenously administered ^13^C-labeled methacetin is metabolized exclusively via the hepatic microsomal localized hemoprotein enzyme 1A2 (CYP1A2) of the cytochrome P450 group [32,33,34]. Metabolites of methacetin are paracetamol at subtherapeutic concentrations and pulmonary eliminable ^13^CO_2_. The LiMAx^®^ value is calculated based on the maximal change in the isotope ratio of ^13^CO_2_ and ^12^CO_2_ compared to the individual baseline (delta over baseline (DOB)). With an elevation of hepatic metabolic efficiency, more ^13^CO_2_ is exhaled. As a result, DOB and LiMAx^®^ values increase [25,34].

Values > 315 μg/kg/h define the normal range. Between 314 µg/kg/h and 140 µg/kg/h, the hepatic metabolic activity is impaired. A LiMAx^®^ value <140 µg/kg/h is associated with a severely impaired hepatic function [35,36]. An increase in LiMAx^®^ levels may indicate the early recovery of liver function before the static liver parameters start to normalize [30,37].

CytoSorb^®^ is a high-tech polymer adsorber with a surface area >45,000 m^2^. It has a strong binding capacity for a variety of predominantly hydrophobic molecules with a molecular weight <55 kDa, such as pro- and anti-inflammatory mediators, immune response-inducing damage-associated molecular patterns (DAMPs) or pathogen-associated molecular patterns (PAMPs), and pharmacological substances (e.g., anticoagulants and psychotropic drugs) [38,39,40,41,42,43,44].

Previous research addressed the possible effects of adsorption of various DAMPs, PAMPs, pro- and anti-inflammatory cytokines. Hemoadsorption with CytoSorb^®^ equally reduces equally elevated levels of pro- and anti-inflammatory cytokines in sepsis in a concentration-dependent manner. Thereover this therapeutic principle does not affect a specific pathway of the immunological network [42,45]. The goals are to rebalance the immunological system [46], to downregulate the excessive, life-threatening hyperinflammation (“cytokine storm”) [42,45,47,48], and to alleviate capillary leakage and vasoplegia [49]. CytoSorb^®^ can potentially decrease mortality [49,50,51,52]. In case of a liver failure, additional removal of bilirubin or bile acids could be an important therapeutic option [38,41,53,54,55].

This prospective clinical investigation evaluated dynamic and static liver function in patients with sepsis or septic shock, who received hemoadsorption with CytoSorb^®^ to mitigate or accommodate exaggerated and dysregulated immunological responses (“shock reversal”).

## 2. Materials and Methods

### 2.1. Ethics

The ethics review board of the medical faculty of the Ruhr University Bochum, located in Bad Oeynhausen, approved the study protocol (file number 2019-440). The research project is called “Change of microcirculation and liver function under CytoSorb^®^ in patients with sepsis and septic shock” (MiHaS-Study). The study is listed in the German Clinical Trials Registry (DRKS00017211). Before including the participants in the study, the patients, the respective caregivers or independent consultants gave their written informed consent.

### 2.2. Study Design

In this prospective, monocentered, investigator-initiated, observational study, patients with sepsis or septic shock [56] and an indication for renal replacement therapy with adjuvant hemoadsorption with CytoSorb^®^ were included between March 2020 and March 2021 (Figure 1). Exclusion criteria were pregnancy, breastfeeding, age < 18 years, allergy to acetaminophen or methacetin, intravenous drug abuse, advanced liver disease (e.g., chronic hepatitis, hepatocellular carcinoma, liver metastases, severe cirrhosis (Child–Pugh C)), prior organ transplantation, and HIV infection. A surgically uncontrollable focus or lack of informed consent also led to exclusion. Furthermore, patients with concurrent use of substances metabolized by cytochrome P450 1A2, e.g., mexiletine, propafenone [57] ciprofloxacin [58,59], imipramine and amitriptyline [60], were excluded. Patients receiving inhaled sedation with isoflurane were not considered due to suspected interference with the LiMAx^®^ measurement [61].

### 2.3. Study Protocol

All patients received individualized standard sepsis therapy according to current guidelines [62] and renal replacement therapy (continuous venovenous hemodialysis (CVVHD)) combined with hemoadsorption according to internal departmental specifications. The duration of CytoSorb^®^ therapy was terminated after the individually calculated amount of blood purified (ABP) was reached [52]. On admission to the ICU, the Acute Physiology and Chronic Health Evaluation II score (APACHE II) was determined [63]. The dynamic liver test LiMAx^®^ was performed at four defined points of time:

T_1_: Preadsorber: (−4 h) before connection to hemoadsorption with CytoSorb^®^.T_2_: 24 h after the start of CytoSorb^®^ therapy.T_3_: 48 h after the start of CytoSorb^®^ therapy.T_4_: Postadsorber: 24 h after completion of CytoSorb^®^ therapy.

In addition, the following parameters were collected at T_1_ to T_4_ as part of routine clinical practice: sequential organ failure assessment score (SOFA) [64], blood gas analysis, inflammatory and liver parameters, hemodynamic and respiratory parameters, and current medication. These datasets were recorded using the patient information system (Integrated Care Manager, Draeger, Lübeck, Germany).

### 2.4. LiMAx^®^ Test

By using the Flip^®^ Analyzer (Humedics GmbH, Berlin, Germany) the LiMAx^®^ test was performed. Then, 2 mg/kg body weight of ^13^C-labeled methacetin was injected intravenously and metabolized to paracetamol and ^13^CO_2_. The DOB was measured and quantified over 60 min [25,34]. Expiratory gas from mechanically ventilated patients was delivered to the ^13^CO_2_ detector via a special adapter. To avoid interactions (recirculation, altered kinetics) and to create stable measurement conditions, renal replacement therapy with hemoadsorption was interrupted during measurements. In addition, the infusion regime, ventilation parameters, and inspiratory oxygen concentration (FiO_2_) [65] were kept nearly stable during the measurements. Due to pharmacokinetic reasons, the time interval between two LiMAx^®^ tests was at least 24 h.

### 2.5. CytoSorb^®^

Renal replacement therapy was always indicated as CVVHD based on relevant guidelines [66]. The multifiltrate Ci-Ca^®^ (Fresenius AG, Bad Homburg, Germany) was used. For an adjunctive hemoadsorption, a CytoSorb^®^ adsorber was integrated into the pre-filter position according to individual indication by the intensivist, in accordance with manufacturer recommendations and internal hospital standards. The adsorber was replaced after 8–24 h, depending on the clinical course of the disease (vasopressor dose, stabilization, inflammation parameters). Following current recommendations, the dialysis dose was between 20 and 30 mL/kg/h [49,66,67]. To maximize the CytoSorb^®^ dose (ABP) a maximum blood flow rate of 200 mL/h under citrate anticoagulation was always targeted, depending on the volume status of the patient and pressure conditions of the CVVHD [52,68].

### 2.6. Definition of Outcomes

The primary endpoints were LiMAx^®^ values during the observation period from T_1_ to T_4_. The secondary endpoints included the course of various static liver parameters, IL-6 levels, and the dose of norepinephrine.

### 2.7. Statistical Analysis

Data were collected electronically using Microsoft Excel (version 16.53; Microsoft Corp. Redmont, WA, USA). All statistical analyses were performed with the program R, version 4.0.5 (https://www.r-project.org (accessed on 5 April 2021)). 

Data were tested for normal distribution with the Shapiro–Wilk test and presented as mean ± standard deviation plus interquartile range (25th and 75th percentiles) or median plus interquartile range (25th and 75th percentiles), whichever was appropriate. 

Group differences (sepsis vs. septic shock) were evaluated with appropriate tests depending on the scale and data distribution: Fisher’s exact test (categorical data), T-test for unrelated samples (normal distribution in all 4 points of measurement in both groups), or Mann–Whitney test. The *p*-values were corrected according to Bonferroni. A two-sided *p*-value < 0.0125 was statistically significant.

Subgroups (sepsis or septic shock) were analyzed by using Fisher’s exact test (categorical data), paired-samples T-test (normal distribution in all 4 points of measurement), or Wilcoxon signed-rank test. A sliding baseline was chosen for the four points of measurement, comparison T_1–2_, T_2–3_, and T_3–4_. A two-sided *p*-value < 0.0167 after α-adjustment by using the Bonferroni correction was set as the statistical threshold.

## 3. Results

### 3.1. Demographic Values

A total of 21 patients (12 female and 9 male) were recruited between March 2020 and March 2021 (Figure 1). The median age was 74 years. BMI was 26.05 ± 6.46. The septic foci were mainly abdominal (*n* = 15) with 71%, pulmonary (*n* = 3), or other parts (*n* = 3). No significant differences in baseline characteristics between sepsis and septic shock were found, except for lactate, which was significantly lower in sepsis (*p* = 0.00002) (Table 1).

A total of 67% (*n* = 14) of patients developed a septic shock. APACHE-II and SOFA scores at T_1_ were: 31.19 ± 4.19 and 14, respectively, *p* = 0.5 and *p* = 0.28. Patients with septic shock had a 15% higher SOFA score than patients with sepsis. A total of 95% (*n* = 20) of patients were mechanically ventilated. Parameters of hemoadsorption showed no differences. The achieved ABP was 13.23 ± 1.71 L/kg for the total cohort (sepsis: 13.52 ± 1.05 L/kg; septic shock: 13.08 ± 1.98 L/kg). 

LOS in ICU was 24.38 ± 15.67 days (range 5 to 59 days). All patients with sepsis survived 90 days. A total of 57% (*n* = 8) of patients with septic shock did not survive the ICU stay, 50% (*n* = 7) of them died between the 10th and 19th day, and 57% (*n* = 8) died within 90 days. The observed 28-day- and 90-day-mortality for the total cohort was 33% (*n* = 7) and 38% (*n* = 8), respectively, *p* = 0.032 and *p* = 0.018. ICU mortality was equal to 90-day-mortality. In summary, every patient, who had been transferred from the ICU to periphery wards, survived.

### 3.2. Hepatic Dysfunction

#### 3.2.1. Dynamic Liver Test: LiMAx^®^

Dynamic liver function was significantly impaired at T_1_ for both groups (*p* = 0.276). There were no differences between sepsis and septic shock (Table 2 and Table 3, Figure 2). Preadsorber (T_1_), 38% (*n* = 8) of all patients had limited liver injury, another 57% (*n* = 12) had a severe liver injury and 5% (*n* = 1) had a normal liver function according to LiMAx^®^ (Table 3). 

The following results refer to the cohort with sepsis (Table 2 and Table 3, Figure 2): 

A total of 43% (*n* = 3) of participants showed a severe liver damage and 43% (*n* = 3) an impaired hepatic function according to LiMAx^®^ (Table 3). The mean values of LiMAx^®^ decreased from T_1_ to T_2_ by 34% (*p* = 0.148) and tended to recover slowly but not significantly after 48 h of hemoadsorption (T_3_: 130.86 ± 66.12 μg/kg/h; *p* = 0.261). From T_3_ to T_4_ the mean values of LiMAx^®^ recovered significantly by 69% (*p* = 0.003) and exceeded the starting point (T_1_). 

The following results refer to the cohort of septic shock (Table 2 and Table 3, Figure 2):

A total of 64% (*n* = 9) of patients had a severely impaired liver function and 36% (*n* = 5) had an impaired liver function. There was a significant deterioration in the mean value of LiMAx^®^ from T_1_ to T_2_ by 45% (*p* = 0.003). After 48 h under hemoadsorption (T_3_), the dynamic liver function tended to improve from T_2_ to T_3_ (*p* = 0.025). This mean value recovered significantly from T_3_ to T_4_ by 92% (*p* = 0.001) 24 h after the end of CytoSorb^®^ therapy. T_4_ had the highest LiMAx^®^ results (T_4_: 188.36 µg/kg/h).

#### 3.2.2. Static Liver Parameters

There were no significant differences in static liver parameters observed between sepsis and septic shock at the four points of time (Table 2). The cumulative incidence of liver dysfunction varied depending on the different definitions. According to the third international Consensus definition (“Sepsis-3”) [56], liver dysfunction is based only on serum bilirubin concentration and hyperbilirubinemia starts >1.2 mg [56]. Related to the “Sepsis-3”, in the overall cohort, 14% (*n* = 3) of patients showed hepatic insufficiency (Table 3). When referring to ALT only 38% (*n* = 8) of all patients had impaired liver function at T_1_, and elevated ALT levels were still observed in six patients at T_4_ (Table 3). 

The following results refer to the cohort with sepsis (Table 2 and Table 3): INR increased significantly from T_1_ (1.24) to T_2_ (1.38) by 11% (*p* = 0.0002), followed by a non-significant decrease to T_4_ (1.04). There were no significant changes in other static liver parameters. The median of ALT were below the pathological reference range (<50 U/L) and constantly declined over time from T_1_ to T_4_ by 38%. Bilirubin levels decreased (T_1_: 0.43 mg/dL; T_4_: 0.37 mg/dL) during and after CytoSorb^®^ therapy (T_3_; T_4_).

The following results relate to patients with septic shock (Table 2 and Table 3):

From T_1_ to T_2_, the INR increased significantly by 26% (*p* = 0.006), followed by a slow decrease up to T_4_. No significant changes were observed over time for ALT, Bilirubin, and AP. At the starting point (T_1_) ALT was not pathologically elevated based on median (T_1_: 33.5 U/L). ALT had an upward (T_2_, T_3_) followed by a downward (T_4_) course during the observation period. The median value at T_4_ was slightly above the normal range, and the 75th percentile was significantly elevated from T_1_ to T_4_. A total of 43% (*n* = 6) of the patients had pathologically elevated ALT levels at T_1_ and 36% (*n* = 5) at T_4_. Bilirubin was within the normal range related to the median and IQR at all points of time. A total of 21% (*n* = 3) of the patients had hyperbilirubinemia before the start and after the end of hemoadsorption. AP showed an unremarkable course in the normal range from T_1_ to T_3_ and increased after hemoadsorption with CytoSorb^®^ from T_3_ to T_4_ by about 35% in the median.

### 3.3. Shock Reversal

At T_2_ norepinephrine dose was 133% higher in patients with septic shock than in patients with sepsis (*p* = 0.007) (Table 4). Norepinephrine doses in both groups converged during the course. Vasopressor support decreased continuously from T_2_ to T_4_. The median level of IL-6 was pathologically elevated in both groups at all points of time. The IL-6 concentration tended to be higher in patients with septic shock at all points of measurement (Table 4 and Figure 3). 

The following results relate to the group of sepsis (Table 4, Figure 3): 

A significant IL-6 reduction was detecTable 24 h after initiation of CytoSorb^®^ from T_1_ to T_2_ by about 73% (*p* = 0.016) and after T_2_ the values dropped continuously. All participants in this group received norepinephrine therapy and norepinephrine initially increased until T_2_. A significant decrease became apparent after 48 h hemoadsorption from T_2_ to T_3_ by 73% (*p* = 0.016).

The following results relate to the cohort with septic shock (Table 4, Figure 3):

There was a reduction in IL-6 from T_1_ to T_2_ by 90% (*p* = 0.0004). Initially, the norepinephrine requirement increased significantly (T_1_ to T_2_) by 154% (*p* = 0.005). After T_2_ a consistent and significant reduction in norepinephrine could be observed (T_2_–T_3_: *p* = 0.0001; T_3_–T_4_: *p* = 0.0006). After the end of hemoadsorption, the dose decreased from T_3_ to T_4_ by 69%.

## 4. Discussion

Sepsis can lead to micro- and macrocirculatory vasculopathy [13,69,70,71]. The reversal of septic shock, i.e., improved microcirculation by control of inflammation [72] with a decrease in different surrogate parameters, such as catecholamine dose or lactate concentration, is the most important goal of septic shock treatment [62,73].

Under hemoadsorption with CytoSorb^®^, hemodynamic stabilization has already been demonstrated by several research projects [74,75,76,77,78,79]. However, an immune modulatory effect of hemoadsorption with CytoSorb^®^ on dynamic and static liver function in sepsis has not yet been investigated in a comparable setting. 

Typical forms of SALD include hypoxic hepatitis, also known as “shock liver” [80,81,82], cholestatic dysfunction [22,83,84,85,86], secondary sclerosing cholangitis and can be caused by hepatotoxic drugs [10,87,88,89]. 

### 4.1. Aspects of Feasibility

In the present prospective study, LiMAx^®^ proved to be safe, feasible and reliable under mechanical ventilation. It also had the target of exploring its diagnostic potential during hemoadsorption with CytoSorb^®^. As shown in our study, this dynamic test is a practical, bedside, noninvasive tool [33] to determine the metabolic capacity of the liver during CytoSorb^®^ in critically ill patients. 

Pausing renal replacement therapy during the whole 60 min of the LiMAx^®^ measurement cycle to exclude any possible interaction or recirculation via CRRT and CytoSorb^®^ represents a significant strength of this study contributing to robust data quality. In contrast, previous investigations had used much shorter interruptions of dialysis (60 s) [25,26]. 

LiMAx^®^ achieved promising results for the assessment of hepatic function in liver resection, in liver transplantation [25,34,90,91], in monitoring antibiotics [27,29,92] and during sepsis [24,28]. The scope of applications may extend to monitor hepatic function during hemoadsorption with CytoSorb^®^.

### 4.2. Decrease in Plasma IL-6 Levels and Shock Reversal

We observed initially high IL-6 levels followed by a rapid decrease after the initiation of CytoSorb^®^ therapy. A hemodynamic stabilization and the reduction in norepinephrine dose occurred 48 h after the start of CytoSorb^®^, especially in septic shock. According to these results, rapid control of inflammation and rapid shock reversal can be assumed in both groups. This is one of the primary therapeutic goals of hemoadsorption by CytoSorb^®^. Other objectives are the prevention or limitation of (multi-) organ failure, improvement of endothelial integrity and reduction in leakages through immunomodulation [76,93,94,95]. Our findings regarding hemodynamic stabilization, the decrease in norepinephrine and IL-6 are in common with other studies [49,74,78,96,97,98,99]. A highly significant parameter for clinical stabilization in sepsis and septic shock is potentially the required dosage of vasopressor, primarily norepinephrine [62,73]. 

IL-6 levels correlate with liver dysfunction [26] and influence hepatic biosynthesis via genetic processes with up- or downregulation of different components of the proteome, e.g., acute phase proteins [100]. Inflammation and high concentrations of cytokines such as IL-6 cause hepatic dysfunction such as hepatic inflammation, activation of proapoptotic signaling cascades, or disruption of hepatobiliary transport [13,86,101,102,103,104]. 

Furthermore, high IL-6 levels can potentially be used as a criterion of indication for a successful treatment with hemoadsorption [41,94,98,105,106]. Nevertheless, the use of IL-6 as a parameter for indication, follow-up and therapeutic success of hemoadsorption is controversially discussed [74,96,98,107]. The recently presented scoring system could be another option for the indication of CytoSorb^®^ [108]. Primarily, our data demonstrated a rapid hemodynamic stabilization and a quick shock reversal, similarly to other studies [49,74,78,97,98]. A rapid achievement of this specific therapy objective might have a liver-protective effect, which is discussed in Section 4.3.

### 4.3. LiMAx^®^ under Hemoadsorption with CytoSorb^®^: Progression and Clinical Benefits

The study has shown very low pathological LiMAx^®^ values at the onset of sepsis and septic shock in the vast majority of patients. Furthermore, our results show maximal impairment of CYP1A2 activity measured with LiMAx^®^ 24 h after the start of CytoSorb^®^ treatment, especially in patients with septic shock and extremely elevated IL-6 levels. Despite rapid shock reversal, a significant hepatic improvement was evident 24 h after the end of CytoSorb^®^. Thus, the SALD recovered significantly at the subcellular level approximately in a median of 106 h (=4.4 days) after the onset of this life-threatening disease. The metabolic function of the liver was still compromised at T_4_. A return to normal biotransformation capacity (>315 μg/kg/h) was not detectable even 4–5 days after the start of therapy.

Kaffarnik et al. came to a similar conclusion: after the onset of sepsis, the LiMAx^®^ value decreased until the 2nd day, followed by a hepatic recovery on the 5th and 10th day [24]. Comparing Kaffarnik et al. and our cohort, the severity of the disease is higher in the current project, as indicated by the SOFA score: 11.4 and 6.9 [24] vs. 14.3 and 13.9 (Table 1) [24].

In general, a LiMAx^®^ value < 100 μg/kg/h indicates critical liver failure or terminal liver failure [109] and predicts mortality with a sensitivity of 100% in sepsis [24]. Study results prove that LiMAx^®^ can detect early SALD [24].

After the onset of sepsis, 95% (*n* = 20) had a reduced CYP 1A2 activity, either an impaired hepatic function (38%, *n* = 8) or a severe liver damage (57%, *n* = 12). A similar study had comparable LiMAx^®^ readouts [24]. The main reason for SALD is considered to be decreased hepatic blood flow due to a hemodynamic destabilization [110]. Additionally, another trigger of SALD appears to be abnormal hepatobiliary transport and impaired biotransformation of detoxification, particularly in the cytochrome P450 enzyme family, based on a rodent model [111]. The compromised detoxification of different Cytochrome P enzymes could be confirmed in patients with sepsis. As a result, the drug metabolism mediated by CYP P450 was reduced [112]. CYP1A2 and CYP2C19 are negatively correlated with levels of IL-6. High Il-6 levels decrease the drug metabolism of CYP1A2. There may exist a higher risk for augmentation of adverse drug reactions and variability in drug response [113]. These findings give evidence that hepatic impairment might be multifactorial. The hypotension, high IL-6 levels, especially in septic shock, followed by biotransformed changes and dysfunction of hepatobiliary transport may cause SALD. It is also an explanation of the low LiMAx^®^ values in this study, which refer to hepatic damage or failure.

We show that the biotransformation capacity of the liver is extremely affected at the onset of sepsis and septic shock. Despite CytoSorb^®^, the hepatic metabolism measured by LiMAx^®^ recovers hesitantly 24 h after the end of hemoadsorption. It is conceivable that a faster decrease in inflammation parameters such as IL-6 and a quick hemodynamic stabilization due to CytoSorb^®^ has a positive impact on the liver, especially on the hepatic metabolism and the functional capacity of CYP1A2. For this purpose, a clinical study with a control group should be designed.

Currently, there is no specific therapy established for SALD, but based on evidence of hepatic impairment and compromised CYP1A2 metabolism, intensive care strategies should strive for liver-protective therapies whenever possible. In clinical practice, surrogate parameters such as IL-6 and vasopressor requirements are commonly used to assess the clinical course of SALD, but they do not fully reflect the conditions at the subcellular level and the impaired biotransformation of detoxification. This fact should be considered when making therapeutic decisions. The LiMAx^®^ can serve as a guide and help to correctly assess the severity of SALD at an early stage. This diagnostic tool could make an important contribution to minimizing additional liver cell damage, e.g., by adjusting and monitoring liver toxic drugs such as antibiotics and antimycotics [24,27,28,29,31,92,114,115].

### 4.4. LiMAx^®^ Is Superior to Assess Dysfunction Compared to Static Hepatic Parameters 

The median values of all static hepatic biomarkers, except INR, remained within the normal range at almost all points of measurement. Individual patients showed elevated static liver parameters (Table 3).

We observed pathological INR with an incidence of up to 57% (*n* = 7) in the total cohort. Elevated INR levels are associated with coagulopathy, a common symptom of sepsis, which occurs in up to 70% of patients with sepsis [116,117,118]. INR is a reliable diagnostic tool for the definition of acute liver failure [119,120,121]. The incidence of hepatic pathological values is about 38% lower with INR (57%, *n* = 12) than with LiMAx^®^ (95%, *n* = 20).

A similar result occurs with ALT. The hepatic integrity parameter was increased in up to 38% (*n* = 8) of the patients in our study. ALT tended to indicate a possible deterioration of liver function, but it does not correlate with the extent of hepatocellular damage [122]. Therefore, it is impossible to deduce the exact hepatic dysfunction or damage based on ALT.

Furthermore, Bilirubin, a biomarker for cholestatic dysfunction, may indicate bile duct impairment [123]. Under treatment with CytoSorb^®^, the serum bilirubin concentration is neither suitable to assess liver function nor to calculate the SOFA or the MELD score, because it does not correctly reflect the hepatic excretory function. Bilirubin levels are falsely determined to be low because CytoSorb^®^ is able to adsorb this molecule irreversibly [38,55].

Our data clearly show that liver function is early and often severely impaired in sepsis and septic shock according to the LiMAx^®^ (Table 2 and Table 3). Our data also confirm the fact that paraclinical static “liver values” are very limited in assessing hepatic function in the early stage of sepsis and septic shock [100]. Moreover, the validity of biochemical parameters could be limited due to an interfering influence of inflammation, tissue damage, or renal failure [87,104].

In summary, static liver parameters do not adequately reflect the hepatic dysfunction and the impaired hepatic metabolism at the onset of sepsis and during hemoadsorption with CytoSorb^®^. It can be assumed that there is a clear advantage of the LiMAx^®^ test in diagnosing SALD, as previously described by Kaffarnik et al. [24,26]. This also applies to hemoadsorption with CytoSorb^®^ as demonstrated in the current research project.

## 5. Limitations

The present study was planned as an observational trial and should provide results regarding the feasibility and interpretability of the dynamic liver function test (LiMAx^®^) under hemoadsorption with CytoSorb^®^ in sepsis and septic shock. The intention was to create an approach for future research and to generate new hypotheses.

The number of patients included is too small to draw definitive conclusions regarding other outcome parameters, e.g., mortality, or to generate clinical guidelines or strategies.

The usability of the LiMAx^®^ test is significantly limited by the personnel, structural and financial efforts. Therefore, the metabolic necessity of a minimum interval of 24 h between two measurements and the breaks in CRRT treatment for the duration of the measurement interval (about 1 h) require extra organizational efforts in order to avoid any interference.

Despite the prospective study design, certain limitations must be considered. All measurements were performed after the onset of sepsis. Earlier comparative values are missing. Longer-term influences of adjuvant hemoadsorption with CytoSorb^®^ on hepatic biotransformation cannot be excluded with certainty because the study period was about 5 days.

## 6. Conclusions and Perspectives

This is the first prospective study to measure LiMAx^®^ in combination with hemoadsorption by CytoSorb^®^ in mechanically ventilated critically ill ICU patients. It appears that the hepatic biotransformation capacity measured by LiMAx^®^ is massively and significantly impaired at the onset of sepsis. A significant hepatic improvement of metabolic capacity occurred 24 h after the end of hemoadsorption but the liver function was still compromised. It is of utmost importance to choose a liver-protective treatment regime to avoid further damage. Hepatic recovery was paramount during rapid shock reversal facilitated by CytoSorb^®^ with a significant decrease in IL-6 and vasopressor support. Whereas static liver parameters do not adequately reflect the hepatic dysfunction and the impaired hepatic metabolism at the onset of sepsis and during hemoadsorption with CytoSorb^®^. In addition, due to bilirubin removal with CytoSorb^®^, common liver function assessments such as the SOFA or MELD score cannot be used. The LiMAx^®^ test can reliably help to assess the severity and the course of hepatic dysfunction at the subcellular level and to make therapeutic decisions faster, individualized, and thus optimized.

## Figures and Tables

**Figure 1 biomedicines-10-02340-f001:**
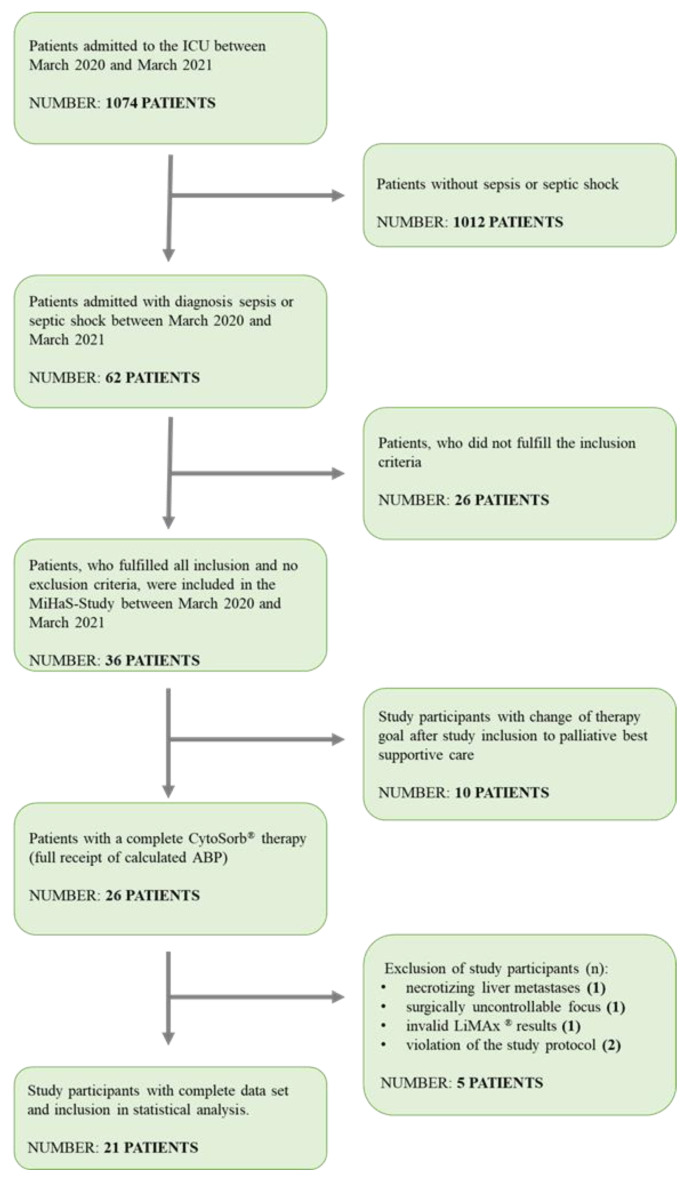
CONSORT statement.

**Figure 2 biomedicines-10-02340-f002:**
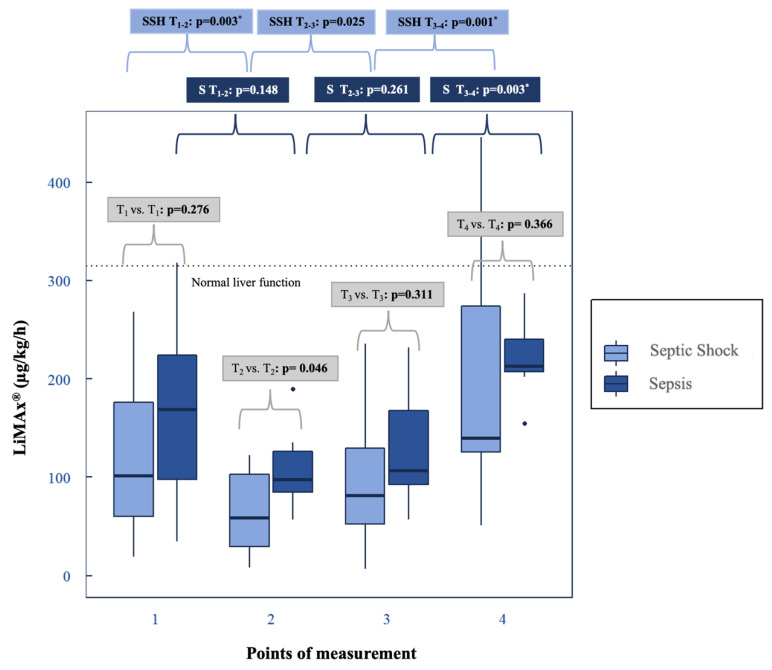
LiMAx^®^: sepsis vs. septic shock and the progression in the subgroups. LiMAx^®^: grey rectangle: *p*-values of septic shock vs. sepsis; dark blue rectangle: *p*-values of the subgroup sepsis (S); light blue rectangle: *p*-values of the subgroup septic shock (SSH); *: statistically significant. Box-whisker plot: bold lines: medians; box plots: 25th to 75th percentiles; box-whisker plots: illustrate median value with interquartile range; outliers are shown as separate points.

**Figure 3 biomedicines-10-02340-f003:**
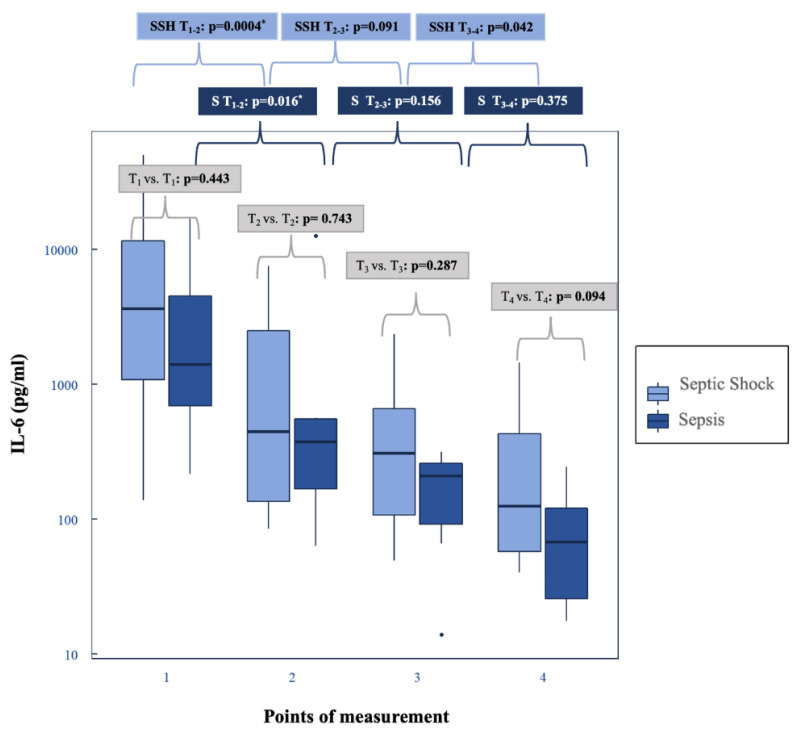
IL-6: sepsis vs. septic shock and the progression in the subgroups. IL-6: LiMAx^®^: grey rectangle: *p*-values of septic shock vs. sepsis; dark blue rectangle: *p*-values of the subgroup sepsis (S); light blue rectangle: *p*-values of the subgroup septic shock (SSH); *: statistically significant. Box-whisker plot: bold lines: medians; box plots: 25th to 75th percentiles; box-whisker plots: illustrate median value with interquartile range; outliers are shown as separate points.

**Table 1 biomedicines-10-02340-t001:** Baseline of demographic and clinical characteristics.

Parameter	Total Cohort	Sepsis	Septic Shock	*p*-Value *
Age (years)	74 (58–80)	60 (54–79.5)	76.5 (60–80.75)	0.28
Gender *n* (%) female male	12 (57%) 9 (43%)	6 (86%) 1 (14%)	6 (42%) 8 (58%)	0.16
Weight (kg)	72.52 ± 19.40	71.14 ± 25.29	73.21 ± 16.80	0.85
Height (cm)	166.57 ± 9.61	163.57 ± 10.00	168.07 ± 9.41	0.34
BMI (kg/m^2^)	26.05 ± 6.46	26.45 ± 9.07	25.85 ± 5.10	0.87
APACHE II score	31.19 ± 4.19	30.43 ± 2.7	31.57 ± 4.82	0.5
Predicted mortality APACHE II (%)	83.73 ± 7.94	83.17 ± 5.72	84.02 ± 9.03	0.8
SOFA score	14 (12–15)	13 (12–14.5)	15 (13.25–15)	0.28
LOS in ICU (days) 28 d observed mortality *n* (%) ICU observed mortality *n* (%) 90 d observed mortality *n* (%)	24.38 ± 15.67 7 (33%) 8 (38%) 8 (38%)	22.29 ± 17.73 0 (0%) 0 (0%) 0 (0%)	25.43 ± 15.13 7 (50%) 8 (57%) 8 (57%)	0.70 0.032 0.018 0.018
Septic focus *n* (%) Abdominal Others: pulmonary pelvinal lower extremity	15 (71%) 6 (29%) 3 (14%) 2 (10%) 1 (5%)	6 (86%) 1 (14%) 1 (14%) 0 0	9 (65%) 5 (35%) 2 (14%) 2 (14%) 1 (7%)	0.34
Duration of CytoSorb^®^ application/patient (h)	82.51 ± 18.02	80.74 ± 24.56	83.39 ± 14.79	0.8
Number of CytoSorb^®^ adsorbers/patient (*n*)	5.10 ± 2.17	4.14 ± 2.12	5.57 ± 2.10	0.17
ABP (L/kg)	13.23 ± 1.71	13.52 ± 1.05	13.08 ± 1.98	0.52
Level of sepsis (Sepsis-3) Sepsis Septic shock	7 (33%) 14 (67%)	0 14	7 0	
Lactate mmol/L	2.27 (1.79–3.99)	1.39 (1.29–1.71)	3.75 (2.4–4.28)	0.00002 #

Legend: Age, height, weight, APACHE II, SOFA, LOS in ICU, Duration of CytoSorb^®^ application and number of CytoSorb^®^ adsorbers: Mean ± SD or median, ( ): IQR, all other values: Absolute number and percentage share, *: Statistical comparison between sepsis and septic shock, #: significant *p*-value: *p* < 0.05.

**Table 2 biomedicines-10-02340-t002:** LiMAx^®^ values and parameters of hepatocellular damage, cholestasis and synthesis in sepsis and septic shock.

Parameter	Point of Measurement		Sepsis	Septic Shock	Sepsis vs. Septic Shock *p*-Value	Sepsis *p*-Value	Septic Shock *p*-Value
**LiMAx^®^** (µg/kg/h)	Preadsorber	T_1_	166.43 ± 99.53 * (97.5–224)	117.36 ± 73.76 * (60–176)	0.276		
	Hemoadsorption (CytoSorb^®^)	T_2_	109.43 ± 43.95 * (84.5–126)	64.86 ± 40.62 * (29.75–102.75)	0.046	0.148	0.003 #
		T_3_	130.86 ± 66.12 * (92.5–167.5)	98.14 ± 68.66 * (52.25–129.75)	0.311	0.261	0.025
	Postadsorber	T_4_	221.43 ± 40.95 * (207–240.5)	188.36 ± 120.09 * (125.25–274.25)	0.366	0.003 #	0.001 #
**ALT (U/L)** (Reference range < 50)	Preadsorber	T_1_	34 * (16.5–53)	33.5 (15.25–161)	0.502		
	Hemoadsorption (CytoSorb^®^)	T_2_	25 * (19–33)	48 (14.25–403.25)	0.412	0.208	0.195
		T_3_	21 * (16–28)	58.5 (16.75–246.75)	0.117	0.025	0.03
	Postadsorber	T_4_	21 * (16.5–30.5)	34 (13.25–181.25)	0.654	0.593	0.055
**Alkaline Phosphatase (U/L)** (Reference range: 40–129)	Preadsorber	T_1_	64 (50–66)	101 * (65.5–145)	0.073		
	Hemoadsorption (CytoSorb^®^)	T_2_	59 (54.5–64.5)	88 (70.25–110.5)	0.062	1	0.124
		T_3_	76 * (59–97)	109 * (73.75–141.5)	0.156	0.059	0.028
	Postadsorber	T_4_	121 * (81.5–166.5)	147.5 (80.25–202)	0.765	0.034	0.019
**Bilirubin (mg/dL)** (Reference range: <1.2)	Preadsorber	T_1_	0.43 * (0.35–0.87)	0.51 (0.47–0.63)	0.737		
	Hemoadsorption (CytoSorb^®^)	T_2_	0.48 * (0.25–0.64)	0.45 (0.37–0.94)	0.391	0.02	0.326
		T_3_	0.36 * (0.24–0.42)	0.56 * (0.42–0.69)	0.015	0.248	0.51
	Postadsorber	T_4_	0.37 * (0.29–0.42)	0.62 * (0.46–1.04)	0.037	0.537	0.091
**INR**	Preadsorber	T_1_	1.24 * (1.21–1.32)	1.28 (1.19–1.45)	0.55		
	Hemoadsorption (CytoSorb^®^)	T_2_	1.38 * (1.33–1.48)	1.61 * (1.45–1.72)	0.062	0.0002 #	0.006#
		T_3_	1.40 * (1.25–1.44)	1.37 * (1.24–1.57)	0.654	0.203	0.03
	Postadsorber	T_4_	1.04 * (1.00–1.08)	1.22 * (1.07–1.29)	0.04	0.002	0.018

Sepsis vs. Septic Shock: Data presented as median (IQR), exception LiMAx^®^: Mean ± SD (normal distribution at all times of measurement),*: Normal distribution; #: Significant *p*-value: *p* < 0.0125. Sepsis and Septic Shock: Sliding baseline (comparison T_1–2_; T_2–3_ and T_3–4_); Data presented as median (IQR), exception LiMAx^®^: Mean ± SD (normal distribution at all times of measurement),*: Normal distribution; #: Significant *p*-value: *p* < 0.0167.

**Table 3 biomedicines-10-02340-t003:** Hepatic impairment or liver failure in sepsis and septic shock.

Parameter	Point of Measurement		Sepsis	Septic Shock
**LiMAx^®^**	Preadsorber	T_1_	3 (43%)	5 (36%)
Impaired liver function	Hemoadsorption (CytoSorb^®^)	T_2_	1 (14%)	0 (0%)
(314–140 µg/kg/h)		T_3_	2 (28%)	3 (21%)
	Postadsorber	T_4_	7 (100%)	5 (36%)
**LiMAx^®^**	Preadsorber	T_1_	3 (43%)	9 (64%)
Severe liver damage	Hemoadsorption (CytoSorb^®^)	T_2_	6 (86%)	14 (100%)
(0–139 µg/kg/h)		T_3_	5 (71%)	11 (79%)
	Postadsorber	T_4_	0 (0%)	7 (50%)
**Elevated ALT**	Preadsorber	T_1_	2 (29%)	6 (43%)
(≥50 U/L)	Hemoadsorption (CytoSorb^®^)	T_2_	1 (14%)	7 (50%)
		T_3_	0 (0%)	7 (50%)
	Postadsorber	T_4_	1 (14%)	5 (36%)
**Elevated alkaline phosphatase**	Preadsorber	T_1_	1 (14%)	5 (36%)
(≥129 U/L)	Hemoadsorption (CytoSorb^®^)	T_2_	1 (14%)	3 (21%)
		T_3_	1 (14%)	4 (29%)
	Postadsorber	T_4_	3 (43%)	8 (57%)
**Hyperbilirubinemia**	Preadsorber	T_1_	0 (0%)	3 (21%)
(≥1.2 mg/dL)	Hemoadsorption (CytoSorb^®^)	T_2_	0 (0%)	3 (21%)
		T_3_	0 (0%)	0 (0%)
	Postadsorber	T_4_	0 (0%)	3 (21%)
**INR** (≥1.5)	Preadsorber	T_1_	0 (0%)	3 (21%)
	Hemoadsorption (CytoSorb^®^)	T_2_	2 (29%)	10 (71%)
		T_3_	1 (14%)	6 (43%)
	Postadsorber	T_4_	0 (0%)	2 (14%)

Legend: total number and percentage (%) of patients.

**Table 4 biomedicines-10-02340-t004:** Parameter of inflammation and shock reversal in sepsis and septic shock.

Parameter	Point of Measurement		Sepsis	Septic Shock	Sepsis vs. Septic Shock *p*-Value	Sepsis *p*-Value	Septic Shock *p*-Value
**Interleukin-6** (pg/mL) (Reference range: <7)	Preadsorber	T_1_	1402 (727.75–5317)	4713 (1089.5–11717.5)	0.443		
	Hemoadsorption (CytoSorb^®^)	T_2_	374.20 (175.15–553.2)	448.35 (135.75–2593.25)	0.743	0.016 #	0.0004 #
		T_3_	210.4 * (97.15–263.90))	329.4 (107.68–668.33)	0.287	0.156	0.091
	Postadsorber	T_4_	68.30 (27.35–139.80)	128.75 (57.58–446.38)	0.094	0.375	0.042
**Norepinephrine dose** (µg/kg/min)	Preadsorber	T_1_	0.09 * (0.07–0.19)	0.22 * (0.11–0.65)	0.19		
	Hemoadsorption (CytoSorb^®^)	T_2_	0.24 * (0.14–0.37)	0.56 * (0.43–0.73)	0.007 #	0.09	0.005 #
		T_3_	0.09 * (0.05–0.10)	0.16 (0.11–0.19)	0.038	0.016 #	0.0001 #
	Postadsorber	T_4_	0.05 * (0.02–0.09)	0.05 (0.02–0.08)	0.97	0.721	0.0006 #

Sepsis vs. Septic Shock: data presented as median (IQR), *: normal distribution; #: significant *p*-value *p* < 0.0125 Sepsis and Septic Shock: sliding baseline (comparison T_1–2_; T_2–3_ and T_3–4_); data presented as median (IQR); *: normal distribution; #: significant *p*-value: *p* < 0.0167.

## Data Availability

The datasets used and analyzed during the current study are available from the corresponding author on reasonable request.

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
