# Peer review of "Immunomodulation by Hemoadsorption—Changes in Hepatic Biotransformation Capacity in Sepsis and Septic Shock: A Prospective Study"

_biomedicines, 2022, doi:10.3390/biomedicines10102340_

Round 1

Reviewer 1 Report

In the present study authors have explored the immunomodulatory effects of hemoadsorption with CytoSorb on liver functions during sepsis. Although the sample size is limited but overall study has been designed aptly and results have been interpreted very well. This study makes a good platform to conduct the study in a large cohort. Besides, authors can rethink to shorten the title of the study as it is little long.

Author Response

Dear Reviewer!

Thank you very much for your constructive feedback and the good rating. 

According to your suggestion, we shortened the title.

Yours  sincerely

Janina Praxenthaler

Reviewer 2 Report

The article covers significant area of sepsis and septic shock. Only few concern:

1. "In Dynamic Liver" Test there are n=1, n=3, authors should increase the n number atleast for n=1.

2. the same n number issue persist for "static liver parameters. "

Please consider to increase the n number and try to be consistent on that to the result section.

Author Response

Dear Reviewer!

Thank you very much for your constructive feedback and the good rating. 

The total number of patients included into this study was 21 resulting in small patient numbers when in comes to percentages (5% (n=1)). 

In case this explanation does not meet your request, please specify your concern.

Yours sincerely 
Janina Praxenthaler